# More frequent intense and long-lived storms dominate the springtime trend in central US rainfall

Zhe Feng[1], L. Ruby Leung[1], Samson Hagos[1], Robert A. Houze[1], Casey D. Burleyson[1] & Karthik Balaguru[1]

The changes in extreme rainfall associated with a warming climate have drawn significant attention in recent years. Mounting evidence shows that sub-daily convective rainfall extremes are increasing faster than the rate of change in the atmospheric precipitable water capacity with a warming climate. However, the response of extreme precipitation depends on the type of storm supported by the meteorological environment. Here using long-term satellite, surface radar and rain-gauge network data and atmospheric reanalyses, we show that the observed increases in springtime total and extreme rainfall in the central United States are dominated by mesoscale convective systems (MCSs), the largest type of convective storm, with increased frequency and intensity of long-lasting MCSs. A strengthening of the southerly low-level jet and its associated moisture transport in the Central/Northern Great Plains, in the overall climatology and particularly on days with long-lasting MCSs, accounts for the changes in the precipitation produced by these storms.

[1] Atmospheric Sciences and Global Change Division, Pacific Northwest National Laboratory, Richland, Washington 99352, USA. Correspondence and requests for materials should be addressed to Z.F. (email: zhe.feng@pnnl.gov).

The observed rate of increase for sub-daily extreme convective precipitation has been found to be as much as doubling of the 7% per °C increase in precipitable water from the Clausius–Clapeyron relation[1,2]. Intensification of short-duration convective storms has significant societal implication as the frequency and magnitude of flash flooding could potentially increase. Mesoscale convective systems (MCSs) are the largest type of convective storm that develop when convection aggregates and induces mesoscale circulation features, which distinguish them from isolated cumulonimbus clouds[3,4]. MCSs have complex convective processes that exhibit different structures of organization[4]. It is currently unknown how MCSs may respond to a warming climate[5].

Over North America, some of the world's most intense MCSs commonly occur east of the Rocky Mountains[6]. The observed frequency and amount of extreme precipitation has been increasing in the recent decades in that region with large areas of the Great Plains and Midwest showing statistically significant upward trends[7–10]. Extreme precipitation is defined here as daily precipitation above a threshold of the 95th percentile of the frequency distribution of precipitation rate[5]. Over the central United States, MCSs contribute 30–70% of the total warm-season precipitation[11,12]. These eastward propagating large convective cloud systems[13,14] also contribute to over half of the extreme 24-h precipitation in this region[15,16]. Up to now, only one study[17] has examined the phenomenological aspects of the observed increases. That study found a statistically significant upward trend during 1908–2009 in extreme precipitation events associated with frontal systems, which are the leading cause of daily extreme precipitation in the central and northern regions of the United States. However, owing to the lack of satellite data in the early period, it is difficult to explicitly identify MCSs associated with frontal systems to be consistent with post-satellite era studies. Therefore, whether the changes in extreme precipitation are more closely linked to changes in mesoscale convection or isolated thunderstorms remains unknown.

Here, we use the high-resolution (1 hourly, 12-km) gridded precipitation data set from the North American Land Data Assimilation System (NLDAS) and hourly rain gauge data from National Climatic Data Center (NCDC). We identify MCSs by characteristics of the spatial and temporal contiguousness of their precipitation field embodied in the NLDAS data set (see Methods). Our method is analogous to others that utilize satellite infrared brightness temperature data to track MCSs[12]. We focus the study on springtime (April-June) because the

central United States seasonal rainfall maximizes during that time, when coupling with baroclinic waves and the Great Plains Low-level Jet (GPLLJ) are most frequent, leading to stronger moisture convergence, more concentrated convective activity, and greater amount of precipitation than in mid-summer[18]. Furthermore, the amount of precipitation produced by successive sequences of MCSs, which have a higher potential for flooding, tend to occur more frequently in spring over the same area[12]. We find that changes in the characteristics of MCSs dominate the observed increase in springtime total and extreme rainfall in the central United States. More precisely, the increased rainfall is primarily associated with increased intensity and frequency of long-lasting MCSs. Surface warming over the Rockies that increased the pressure gradient across the central United States accounts for these changes. The increased pressure gradient strengthened the climatological southerly low-level jet in the Central/Northern Plains, which transports warm air to that region, favouring more intense and longer-lived MCS precipitation.

## Results

**Trends in MCS total precipitation and frequency.** Figure 1 shows the mean climatology and temporal trend of total MCS rainfall east of the Rocky Mountains. During spring season, most MCS precipitation is located in the Southern and Central Plains, consistent with previous studies of the most extreme MCSs[12,19]. The trends in total MCS rainfall are almost exclusively positive in the central United States, with large regions showing a statistically significant upward trend. Many states in the Midwest experienced 0.4–0.8 mm day$^{-1}$ (roughly 20–40%) per decade increases in total MCS rainfall. Spatially, a majority of the increase in MCS rainfall occurs in areas north of the climatological maximum rainfall from MCSs. Examination of the trends in total MCS rainfall during summer (July–August, Supplementary Fig. 1) shows that areas with positive trends are located primarily south of the climatological maximum over the Southern Great Plains, southeastern United States and parts of the Great Lakes region. This suggests that the mechanisms for summer MCS changes are not the same as in spring. If there were a seasonal shift of the MCS precipitation (that is, summer MCSs are occurring earlier in spring), the summer MCS precipitation over the central United States would show a decreasing trend. The lack of decreasing summer MCS precipitation over the central United States suggests that the change is not simply a seasonal shift, but rather that during spring MCSs are occurring more frequently or precipitating more heavily in the Central and Northern Plains.

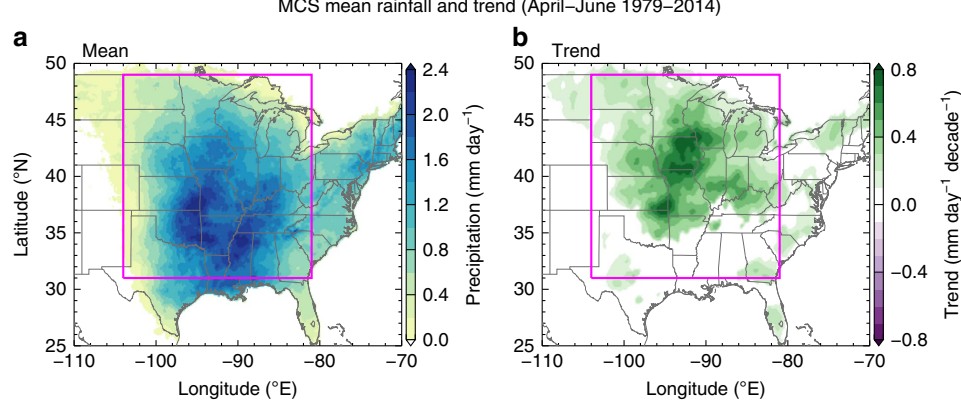

**Figure 1 | Springtime mesoscale convective systems rainfall climatology and trends.** Mesoscale convective system (MCS) (**a**) mean total rainfall and (**b**) total rainfall trend from 1979 to 2014. Total rainfall shown is the accumulated MCS rainfall during April–June divided by the total number of days (91). Only trends with statistical significance above 95% using a two-tailed Student *t*-test are shown. Data within the magenta boxes are used to calculate the trends in Fig. 2.

Annual variations in total, MCS-only and non-MCS daily precipitation in the central United States during spring are shown in Fig. 2a–c. Total precipitation increased at a linear rate of 3% per decade over the period 1979–2014. This increase is attributable primarily to an increase in MCS rainfall, which exhibits a trend of 25% per decade. Non-MCS precipitation shows a slightly decreasing, but statistically insignificant, trend of −2% per decade. Since the incorporation of radar and satellite data in the NLDAS precipitation data set beginning in 1996 (see Methods section), linear trends in the period from 1997 to 2014 for MCS precipitation increased to 28% per decade while non-MCS precipitation decreased at a faster rate of −23% per decade.

To understand the increase in total MCS precipitation, we examined trends in MCS lifetime, frequency and intensity. The average lifetime of MCSs increased at a significant trend of 4% per decade (Fig. 2d). The increase is particularly striking in the frequency of very long-lasting MCSs. The trend of the 95th percentile MCS lifetime is ∼7% per decade. The change in MCS lifetime from 1997 to 2014 exhibits a similar trend.

In accordance with this increase, the MCS precipitation frequency shows a 11% per decade increase, while non-MCS precipitation frequency decreases at −8% per decade. A larger trend during 1997–2014 is found for both MCS and non-MCS precipitation frequency (Fig. 2e,f), consistent with changes in their precipitation amount (Fig. 2b,c). We examined the trends found in Fig. 2 using an independent data set from NCDC hourly rain gauges. Using the time and location of MCSs identified from NLDAS, we separated the NCDC hourly rain gauge data into MCS and non-MCS precipitation (see Methods section). The results show consistent increasing trend in MCS precipitation amount and frequency and decreasing trend for non-MCS precipitation (not shown). The trend magnitudes in the NCDC data set are larger for both MCS and non-MCS precipitation, possibly due to differences between point measurements for the NCDC rain gauges and the spatially interpolated NLDAS data set. Figure 2e,f suggest that when springtime precipitation does occur over the Central United States, there is an increasing tendency for it to be associated with MCSs, which are more efficient at producing larger amounts of rain.

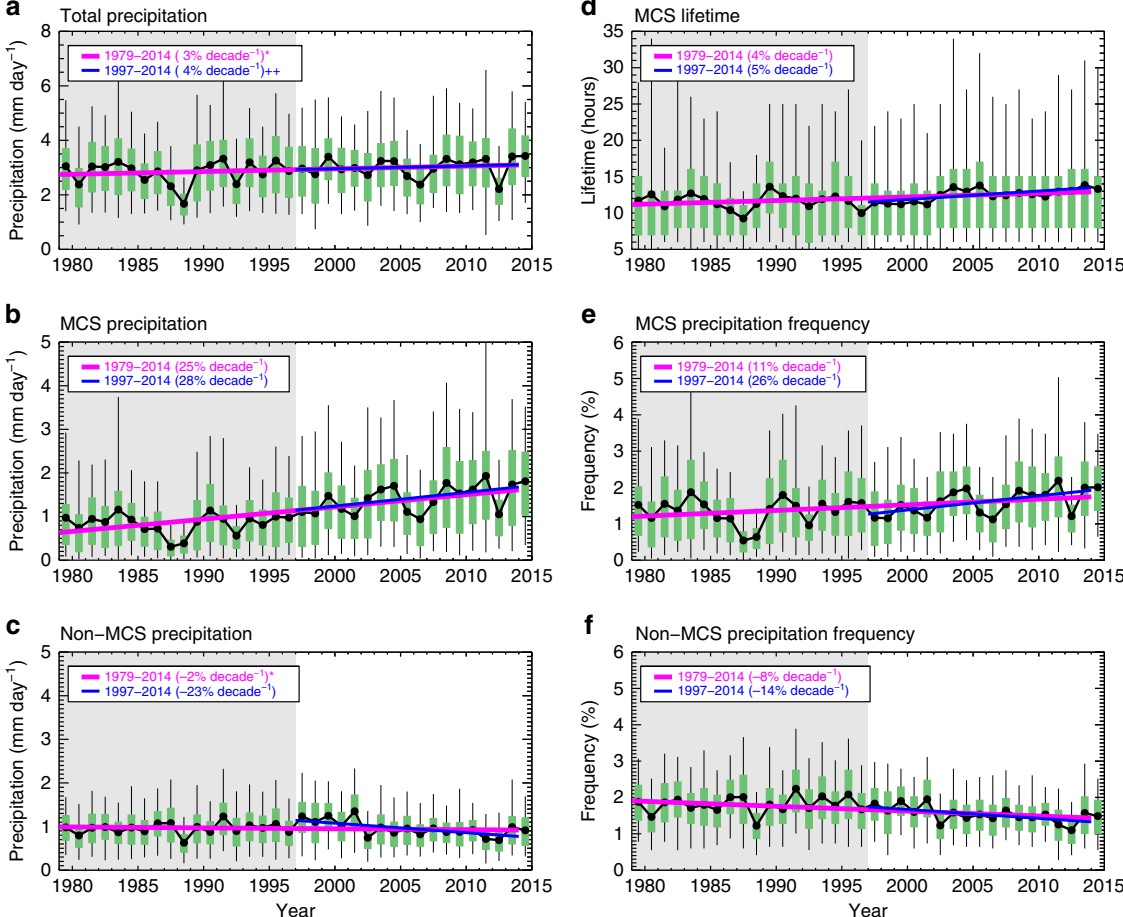

**Figure 2 | Time series and linear trends of precipitation in the Central United States.** Linear trends of April–June (**a**) total precipitation, (**b**) mesoscale convective system (MCS) precipitation, (**c**) non-MCS precipitation, (**d**) MCS lifetime, (**e**) MCS precipitation frequency and (**f**) non-MCS precipitation frequency. Precipitation frequency is calculated on the NASA North American Land Data Assimilation System (NLDAS) native grid-scale by dividing the number of hours of MCS or non-MCS precipitation (hours with rain-rate >1 mm h$^{-1}$) by the total number of hours in each season. Results are calculated within the area of the magenta boxes in Fig. 1. Green boxes show the area between the 25th and 75th percentiles, whiskers denote the 5th and 95th percentiles. Mean values are the black lines with solid circles while regression fit lines to the mean values are the magenta (1979–2014 period) and blue (1997–2014 period) lines. The time period before incorporation of the national radar network data in NLDAS is shaded in gray. Linear trends normalized by the 36-year mean are provided for the entire period (1979–2014) and for the radar period (1997–2014) in the legend of each panel. All trends are statistically significant at 95% confidence interval with a two-tailed Student t-test, except those marked with *which are only significant at 90%. Those marked with + + are not significant at 90%. The significance symbols are shown in the legend of each panel.

**Trends in MCS intensity**. To investigate whether changes in the intensity of MCS precipitation (that is, hourly rainfall) also contribute to the observed increase in total MCS rainfall, we examined the NCDC hourly rain gauge data that is separated into MCS and non-MCS precipitation (see Methods section). Figure 3a,b show the trends of extreme MCS precipitation intensity calculated using two methods: the exceedance frequency of the 95th percentile hourly MCS rainfall from the 36-year record at each station and the 95th percentile hourly MCS rainfall value from each 5-year period at each station. Results from both methods show a spatially coherent positive trend of extreme MCS hourly rainfall intensity over the central United States, where total springtime MCS rainfall has been increasing (Fig. 1b). There are more stations showing statistically significant increased exceedance frequency over the Northern Great Plains compared with the 95th percentile rain-rate trends. This is possibly due to larger interannual variability along with smaller sample sizes for determining the latter. The probability density function (PDF) of hourly rain-rate in 5-year windows for all stations with statistically significant trends are shown in Fig. 3c,d. It is evident that moderate to heavy MCS rainfall intensities (5–30 mm h$^{-1}$) have become more frequent in the past 36 years. On the other hand, the 95th percentile hourly rainfall intensities from non-MCS precipitation have consistently decreased across the central United States (not shown). These findings of increased frequency and intensity of extreme hourly MCS rainfall reveal that previously reported increases of heavy rainfall frequency[7,8,10] are dominated by changes in MCS frequency and intensity, at least during the spring season.

**Role of large-scale environment changes**. To examine the role of the large-scale environments connected with the longer lifetimes and increased precipitation intensity of MCSs, we perform composite and trend analysis using the North American Regional Reanalysis (NARR)[20]. We focus on the environment of a subset of MCSs classified as high-precipitation MCSs (HP-MCSs) identified using the NLDAS data set, which we define as MCSs lasting longer than 8 h and have an accumulated rainfall exceeding the median value of all MCSs that have the same lifetime. HP-MCSs account for ~80% of the total MCS rainfall. The changes in MCS lifetime mainly occur in these long-lasting MCSs (Fig. 2d), and they are associated with extreme precipitation amounts[15,16]. Environmental variables during the lifetime of HP-MCSs occurring in the central United States are composited for each spring season, and the grid-scale linear trend is calculated from the seasonal NARR composites. The relatively large number of HP-MCSs in each spring season (average: 92, minimum: 50) ensures that the environmental composites are representative of the regional circulation pattern when MCSs occur. The continuous supply of low-level unstable, warm, and moist air is often associated with the GPLLJ, which is one of the most important factors associated with the intensity[21] and longevity of MCSs[22]. The NARR has been shown to represent reasonable GPLLJ structures and associated moisture transport compared with sounding observations in the Southern Great Plains[23]. We also analyzed the large-scale environments using the ERA-Interim reanalysis[24] and found trends consistent with the NARR (Supplementary Fig. 2).

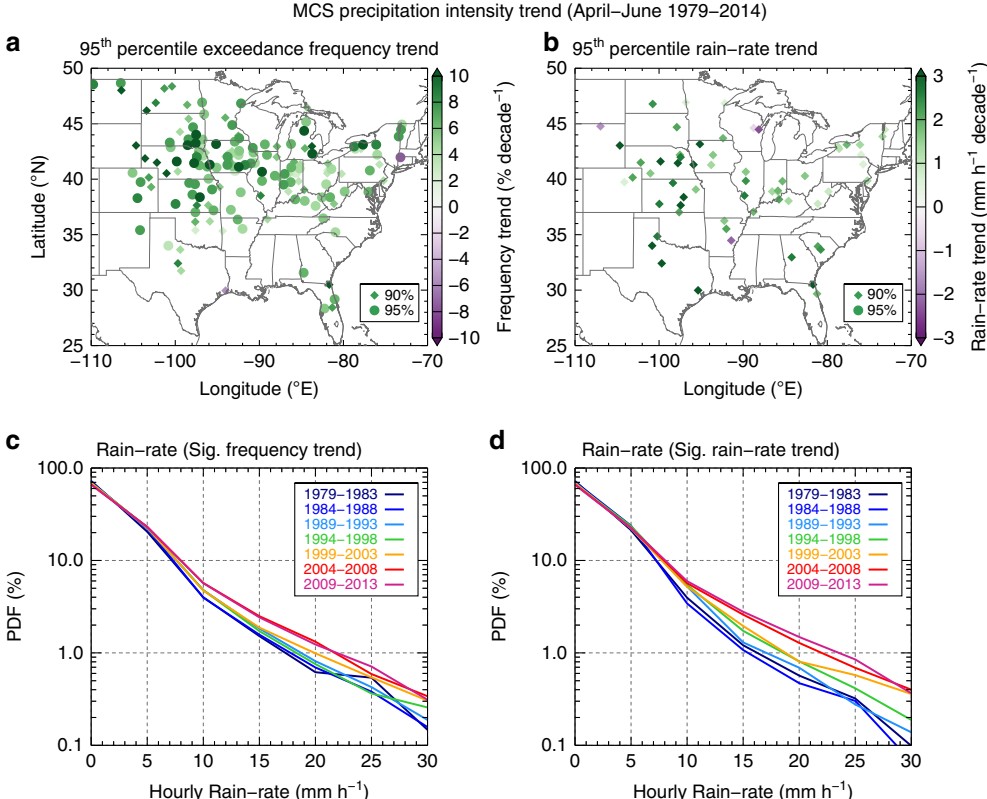

**Figure 3 | Mesoscale convective system extreme precipitation intensity trends.** Trends in mesoscale convective system (MCS) (**a**) exceedance frequency of the 95th percentile hourly rain-rate, (**b**) the actual 95th percentile hourly rain-rate, (**c**) probability density function (PDF) of hourly rain-rates from stations with a significant (Sig.) exceedance frequency trend, and (**d**) PDF of hourly rain-rates from stations with a significant rain-rate trend. The trends are obtained using individual hourly rain gauge data (see the text for more details). Circle and diamond symbols in **a** and **b** show trends that are statistically significant at 95% and 90% with a two-tailed Student t-test, respectively. PDFs in **c** and **d** are constructed using data from all stations with 90% statistically significant trends.

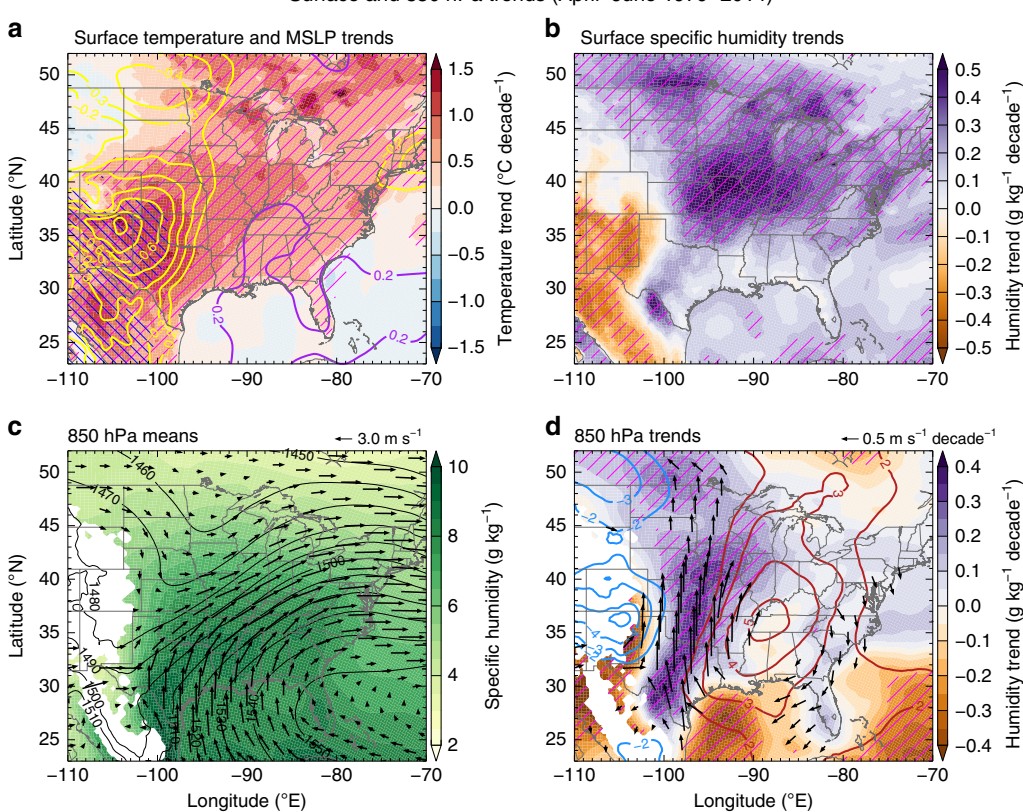

**Figure 4 | Large-scale environment climatology and trends during occurrence of high-precipitation mesoscale convective systems. (a)** Surface temperature (shaded) and mean sea level pressure (MSLP) trends (MSLP contours in 0.1 hPa per decade intervals, purple/yellow contours denote positive/ negative MSLP trends), **(b)** surface specific humidity trends (shaded), **(c)** 850 hPa mean specific humidity (shaded), geopotential height (contours, in 10 m intervals), and wind (arrows), and **(d)** 850 hPa trends in specific humidity (shaded), geopotential height (contours, in 1 m per decade intervals, red/blue contours denote positive/negative geopotential height trends), and wind (arrows, statistically significant at 95%). Grid points with a statistical significance exceeding the 95% confidence interval are marked by **(a)** pink hashes for temperature and blue hashes for MSLP, **(b,d)** purple hashes for specific humidity. Areas with mean surface pressure below 850 hPa are masked out in **c** and **d**.

Figure 4a,b shows the trends in surface air temperature, mean sea-level pressure and specific humidity during observed HP-MCSs for the period 1979–2014. It is evident that HP-MCSs experienced an environment with statistically significant warming at the surface in the central United States. The strongest warming occurs in the Southern Great Plains, just east of the Rocky Mountain foothills. No significant trends are observed over the surrounding oceans. The differential heating between the land and adjacent oceans could shift the mass of the atmosphere towards the southeastern United States[25]. The resulting mean sea-level pressure trend shows a strong decrease near the Rocky Mountain foothills and small increase in the southeast. This spatial pattern strengthens the pressure gradient across the central United States, which in turn enhances the low-level jet. Surface specific humidity also shows a strong positive trend in the Central and Northern Great Plains, consistent with the increase in MCS precipitation (Fig. 1b). These increases combine to promote an increased flux of water vapor into the region of increased MCS activity.

The formation of large MCSs over the Great Plains in springtime is typically associated with baroclinic waves, which favour the development and/or enhancement of the GPLLJ and thus contribute to moisture convergence and vertical motion forcing for the convective systems[18]. GPLLJ events are known to be associated with enhanced precipitation over much of the central and northern Great Plains during spring[26]. Northward migration of the springtime Great Plains maxima of precipitation

frequency and intensity in the recent decades has been linked to the strengthening of the GPLLJ[27]. During spring, the climatological low-pressure system over the Rocky Mountains and high pressure over the southeast and Atlantic Ocean create a broad south/southwesterly moisture transport into the Great Plains (Fig. 4c) that is consistent with the moisture transport by the GPLLJ[23]. The GPLLJ is intermittently increased in strength by the passage of baroclinic waves. The trends of the 850 hPa moisture and circulation (Fig. 4d) are strikingly coherent during spring. Connected with these trends, a statistically significant low-level moistening occurs across a region with a zonal scale of ~1,000 km in the central United States.

The GPLLJ maximizes at night[25]. Comparison of trends between daytime and night time reveals that the nocturnal trends in 850 hPa wind and the northward meridional moisture transport are stronger than the daytime trends (Supplementary Fig. 3). This enhanced transport of low-level moisture through the night favours longer lifetimes of MCSs that peak at night. Moisture convergence at the GPLLJ terminus[18,28] migrates to the Central/Northern Plains along with the northward extension of the GPLLJ. The strengthened GPLLJ transports more moist air to the Northern Great Plains, so MCSs developing in or propagating into that region produce more intense precipitation and last longer, thus facilitating enhancement of total and extreme precipitation, as seen in Figs 2 and 3. Although the aforementioned trends in the large-scale environments are found during days of HP-MCSs, examination of climatological trends

averaged during all days in April–June show consistent patterns, albeit with weaker magnitudes (Supplementary Fig. 4), suggesting that the climatological changes in springtime large-scale environments are at least partially responsible for the increased MCS precipitation.

## Discussion

Our analysis shows that more intense and more frequent longer-lasting MCSs occur along with the trend towards warmer springtime over the central United States. This result has significant implications regarding future extreme rainfall occurrence in this convectively dominated region and season and in other regions frequented by MCSs. Climate model projections suggest that low-level moisture flux will further intensify in a warming climate[25,28]. However, most climate models do not yet have an adequate way of representing MCSs[29,30]. The advent of high-resolution climate models with convection permitting/resolving capabilities should provide a useful tool to improve understanding of the processes controlling the maintenance and characteristics of MCSs and the dynamical feedback related to enhanced boundary layer moisture supply under future warming. This study underscores the urgent need for these improvements.

## Methods

**Data.** The gridded precipitation data over North America were obtained from the NASA North American Land Data Assimilation System (NLDAS) meteorological forcing data set (http://ldas.gsfc.nasa.gov/nldas/NLDAS2forcing.php#AppendixC, available from 1979 to 2014). The NLDAS data set utilizes a combination of ground-based rain gauges, radar, and satellite observations over the United States to produce a 1-hourly 12-km resolution precipitation data set. The hourly precipitation data from rain gauge stations are obtained from the National Climatic Data Center (NCDC, Data Set 3240, https://gis.ncdc.noaa.gov/geoportal/catalog/search/resource/details.page?id=gov.noaa.ncdc:C00313, available from 1948 to 2013). Of the ~2,600 NCDC hourly precipitation stations, 418 sites (east of the Rockies starting at 110°W) were retained. These records have less than 10% missing data between 1979 and 2013. A globally-merged half-hourly 4-km satellite IR brightness temperature data set (http://disc.sci.gsfc.nasa.gov/precipitation/data-holdings/Globally_merged_IR.shtml, available 2000–2014) is used to track cloud systems and identify mesoscale convective systems (MCSs) based on cloud size and duration[12].

**Identifying MCS by precipitation feature.** The 15-year satellite MCS database was used to develop a precipitation feature (PF) algorithm based on NLDAS precipitation data alone by deriving the precipitation characteristics of MCSs. The PF algorithm was then used to construct an independent MCS database, which was validated against the satellite MCS database. The MCS database constructed by the PF algorithm provides consistent MCS rainfall compared with the satellite MCS database, with a majority of the normalized monthly MCS rainfall bias within 15% (see more detail description in 'Developing the precipitation feature method'). The PF algorithm was then applied to the full NLDAS data set to construct a 36-year MCS database from 1979 to 2014. The days with land-falling tropical cyclones in the United States are excluded from the analysis to remove any potential impact of tropical cyclones on precipitation. After MCSs are identified, the total hourly precipitation is separated into MCS and non-MCS (isolated convection or weak stratiform) precipitation at each grid point.

**Developing the precipitation feature method.** The satellite IR brightness temperature ($T_{IR}$) data set (http://disc.sci.gsfc.nasa.gov/precipitation/data-holdings/Globally_merged_IR.shtml, available 2000–2014) is used to track individual cloud systems[31] and identify MCSs based on the size and lifetime of their contiguous cold cloud shield (connected satellite pixels with $T_{IR} < 241$ K). A cloud system with a cold cloud shield area exceeding $6 \times 10^4$ km$^2$ that persists at least 4 h is defined as an MCS[12,32]. The precipitation associated with each MCS is obtained by mapping the NLDAS precipitation onto the MCS cold cloud shield. Supplementary Fig. 5 shows an example of a 24-h long MCS and its associated precipitation tracked by satellite. A PF is defined as a contiguous area with NLDAS pixel-level rain-rate greater than 1 mm h$^{-1}$. Note that the PF and cold cloud shield associated with the MCS have reasonable correspondence in the spatial and temporal evolution throughout most of the life cycle of the MCS. Such correspondence has been explored in previous works using a combination of satellite-based cold cloud shields and ground-based radar and/or rain gauge data set to study the structure and evolution of MCSs[31–35]. For each MCS tracked by the satellite cold cloud shield, the corresponding PF maximum area, maximum areal mean rain-rate, and maximum skewness of the pixel-level rain-rates during its lifetime are obtained.

The PDF of these parameters as a function of MCS lifetime are shown in Supplementary Fig. 6. The rationale for choosing these additional parameters is twofold: physically, precipitation produced by MCSs should exceed a certain size and intensity during their mature stage, with intense convective precipitation elements within the PF that creates positive skewness in the rain-rate PDF; and the characteristics of PFs associated with MCSs clearly show that as MCS lifetime increases, their PF size and intensity also increase (Supplementary Fig. 6). These parameters form the basis for identifying MCSs using the PF data set alone.

To develop a PF algorithm to identify MCSs, the same feature-tracking algorithm derived from analysis of satellite data is adapted to the NLDAS precipitation data set to track PFs rather than cold cloud shields. A PF database is then constructed from the tracked PFs to identify MCSs. Consistent with the satellite definition, we require an MCS to have a PF major axis length exceeding 200 km and persist for at least 4 h. In addition, a tracked PF's maximum area, maximum areal mean rain-rate, and maximum rain-rate skewness must exceed the least square fit values for its corresponding lifetime to be considered as an MCS. In other words, in addition to the minimum size and duration criteria, these three lifetime-integrated PF parameters must be satisfied for an object to be considered an MCS. The least square fit values (thick solid colour lines in Supplementary Fig. 6) are obtained by regressing the PF lifetime ($x$ axis) and a particular percentile value of these three parameters ($y$ axis).

The procedures for obtaining these regression values begin with identification of the same satellite-based MCSs using PF alone. First, a series of percentile values ranging from 5th to 15th percentiles for the three PF parameters are systematically tested to obtain various regression lines. For each regression line combination, the PF-based MCS database is compared with the satellite-based MCS database, focusing on the spatial pattern of MCS accumulated rainfall. A 2-day MCS accumulated rainfall map is used to compute skill scores. Supplementary Fig. 7 shows an example of a comparison with high skill score for the PF-identified MCSs. Over 15 warm-seasons, the PF parameters giving the highest Heidke Skill Score[36] (0.660) and Symmetric Extreme Dependency Score[37] (0.734) on the grid-level 2-day accumulated MCS rainfall map (perfect score is equal to 1) are as follows: the 10th percentile values for maximum area, the 5th percentile values for maximum areal mean rain-rate, and the 5th percentile values for maximum rain-rate skewness. The regression lines for these percentile values are given by the thick solid colour lines in Supplementary Fig. 6. Note that the changes in the overall skill scores within the tested fit percentiles (5th–15th) are within 3.5%, suggesting that the accuracy of the PF-based MCS identification method is not sensitive to the exact choice of the parameters.

We note that the PF-based MCS definitions used here are loose in the sense that any mesoscale (or larger) and intense contiguous rainfall feature exceeding the aforementioned PF parameter thresholds is considered an MCS. Some large weather systems producing widespread (for example, larger than 1,000 km scale) and/or long lasting rainfall (for example, longer than 24 h) are sometimes referred to as synoptic events[17], but individual MCSs do play an important role in producing heavy precipitation associated with these events[7,15]. Thus, sometimes multiple MCSs occurring near a synoptic disturbance could be identified as a single MCS (reflected in the maximum PF length scale >1,000 km and lifetime >24 h) by the PF algorithm due to their spatially and temporally continuous rainfall. This behaviour does not affect the findings in this study, as each of the entities, combined or separated, contribute to net rainfall from MCSs and are forced under similar environments.

Supplementary Fig. 8 shows the distribution of the normalized monthly mean MCS accumulated rainfall error between the satellite-based MCS database and PF-based MCS database:

$$\text{Error} = \overline{\left( \frac{P_{Sat} - P_{PF}}{P_{Sat}} \right)} \qquad (1)$$

where $P_{Sat}$ and $P_{PF}$ are grid-level monthly accumulated MCS rainfall from satellite-based and PF-based MCS database, respectively. Results show that majority of the monthly rainfall errors are smaller than ±15% and are evenly distributed around 0, suggesting that the PF-based MCS database does not have systematic biases compared with the satellite-based MCS database. Over all 15 warm seasons, the spatial distribution of the amount and percentage of MCS rainfall between the satellite-based and PF-based MCS databases are shown in Supplementary Fig. 9. On average, MCSs account for 40–60% of the warm season total rainfall in the central United States from both databases, consistent with previous studies[11,12]. Their spatial distributions also agree quite well (Supplementary Fig. 9).

**Adjusting for changes in NLDAS data set before radar era.** The PF parameters obtained in Supplementary Fig. 6 are then applied to the entire 35-year NLDAS data set during the warm-season to construct the long-term MCS database. A change in the NLDAS precipitation data set was introduced in 1996 when radar and satellite precipitation products were incorporated into NLDAS (for details see http://ldas.gsfc.nasa.gov/nldas/NLDAS2forcing.php#AppendixC). The hourly radar/satellite precipitation products are divided by their respective daily totals to create hourly temporal disaggregation weights to represent proportion of the daily total precipitation that fell within each hour. These hourly weights are then multiplied by the daily gauge-only Climate Prediction Center (CPC) precipitation analysis to obtain the hourly NLDAS precipitation fields. Before 1996, the CPC daily contiguous United States (CONUS) gauge data[38] with 1/8th-degree

PRISM-adjusted analysis is used[39]. As a result, both the PF area and hourly areal mean rain-rate show a noticeable shift around 1996. In general, PF area consistently decreases and areal mean hourly rain-rate consistently increases (Supplementary Fig. 10).

To account for this change, the PF parameter fitting functions derived using data after 1996 (Supplementary Fig. 6) are adjusted. This is accomplished by the following procedures: first, using the data from 1996 to 2000, we obtain the cumulative distribution function (CDF) of the three parameters in Supplementary Fig. 6 for all PFs, including those that do not meet the definition of an MCS. From those CDFs, we find the corresponding percentile values where each of the MCS parameter thresholds fall on the CDFs. Next, we construct similar CDFs of the three parameters using all PFs from 1991 to 1995. We then apply the same percentile values found from data between 1996 and 2000 to the CDFs constructed from all PFs from 1991 to 1995 to obtain adjusted PF thresholds. We perform least square fits on the adjusted PF thresholds to obtain new MCS parameters for pre-radar period (Supplementary Fig. 11). Finally, we apply the adjusted MCS parameters to data from 1979 to 1995 to identify MCSs during that period.

We assume that MCS properties do not change significantly between the two 5-year periods (1991–1995 and 1996–2000). In general, for NLDAS data before1996, the MCS definition is adjusted to have larger areal extent, smaller areal mean rain-rate, and similar pixel-level rain-rate skewness compared with data after 1996. Note that this change does not affect the calculation of accumulated rainfall on temporal scales of one day and longer (Fig. 2), as the daily summation of NLDAS hourly precipitation will exactly reproduce the original CPC daily CONUS gauge data[39], from which the NLDAS was produced. However, it is not appropriate to estimate trends in the hourly precipitation values using NLDAS. Instead, the NCDC rain gauge data is used to calculate trends in hourly MCS precipitation intensity. The NLDAS identified MCS time and location is applied to individual NCDC station data to separate MCS and non-MCS precipitation. PDFs of hourly precipitation values are constructed in 5-year intervals at each station and then the PDFs are used to estimate trends in the extreme percentiles.

**Data availability.** The MCS database produced in this study, along with all relevant data presented are available from the authors upon request.

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

## Acknowledgements

This research was supported by the US Department of Energy, Office of Science, Biological and Environmental Research as part of the Regional and Global Climate Modeling Program. Pacific Northwest National Laboratory is operated by Battelle for the US Department of Energy under Contract DE-AC06-76RLO1830. We thank three anonymous reviewers for providing constructive comments in improving this paper.

## Author contributions

L.R.L. conceived the main idea and provided general direction of this work; S.H. provided crucial guidance in the analysis of the data sets; R.A.H., Jr., provided guidance in the interpretation of the large-scale environment changes. C.D.B. processed the satellite IR brightness temperature data. K.B. provided the list of land falling tropical cyclone dates in the United States.

## Additional information

**Publisher's note**: 

