## [Peer Review File · Nature Communications]

Reviewers' comments:

Reviewer #1 (Remarks to the Author):

I have the following three major comments about the manuscript

1. I find the methods description in the supplementary information somewhat confusing and inadequate. I should be able to reproduce their analysis with this information. There is critical detail missing that prevents a full understanding. For example, I am not sure of the meaning of the least squares fit lines in Fig. S4. What is being regressed against what? It certainly is not a fit to the other data being displayed in that figure. Is this a fit to a particular percentile value for each duration bin?

2. The differences in PF characteristics around the change point of 1996 are surprisingly large (of the order of a factor of 2) and somewhat alarming, raising questions about the adequacy of the sub-daily precipitation. I have concerns about the validity of use of this dataset, even if adjustments are made, particularly in regard to trend analysis. These differences are quite large compared to the trend magnitudes.

3. The statement "Up to now, no study has examined the phenomenological aspects of the observed increases" is not true, at least not without some substantial caveats. I led a rather extensive study of the phenomenology of extreme precipitation increases. This is published as

Kunkel, K.E., D.R. Easterling, D.A.R. Kristovich, B. Gleason, L. Stoecker, and R. Smith, 2012: Meteorological causes of the secular variations in observed extreme precipitation events for the conterminous United States. *J. Hydromet.*, 13, 1131-1141.

In the above study, we found that fronts were principally responsible for the upward trends in extreme precipitation events in the central part of the U.S. On the surface, this is a quite different conclusion. The apparent differences between our findings and those of this study probably derive from the following:

Definition of extremes. The authors here are using the 95th percentile of daily rainfall. That is actually not "extreme" and the authors should use another word, such as "heavy". In my study, we used the 5-yr return period threshold, which is more than an order of magnitude more extreme.

Definition of MCS. My study extended back to the early part of the 20th Century, well before any satellite information could inform that study. We were restricted by what could be identified from surface temperature and precipitation patterns, daily weather maps, and 20th Century reanalysis (before 1948). We categorized the meteorology according to what is likely the primary forcing. If an MCS-like precipitation feature were along a warm or cold front of an extratropical cyclone, we considered the frontal convergence to be the primary cause, although (of course) convective instability could greatly enhance vertical motion. Our MCS category was narrowly defined to those precipitation features that were likely (initiated and) driven primarily by convective instability, some distance from any ETC front. Our identification was done by expert judgment, i.e. we for the most part did not use automated tools. This was quite laborious and, although potentially subject to certain biases, it gave us an intimate feel for the phenomenology for these situations.

Our trend analysis was done for a 100+-year period. While we found overall upward trends for frontally-caused events, a substantial portion of that came from increases prior to 1979. Also, our trend analysis was annual. We did not perform a seasonal trend analysis.

Reviewer #2 (Remarks to the Author):

General comments:

In this study, the authors analyze rainfall data for the late spring (April, May, June; hereinafter

AMJ) in the central United States, and aim to identify trends over the period 1979-2014 in the amounts of rainfall produced by mesoscale convective systems (MCSs) in this region. Based on the analysis, they show that particularly in the upper Midwest (Iowa, Wisconsin, etc.) there has been a large increase in MCS-produced rainfall over this time period and that, in turn, this is associated with a stronger Great Plains Low-Level Jet (GPLLJ) during MCSs. The data and analysis methods are generally sound and the manuscript is very clearly written and presented. However, I have one fairly substantive question/comment that may potentially affect the interpretation of the results that I believe the authors should address prior to the manuscript being ready for publication, along with a few minor suggested revisions.

Major comment:

The primary question/concern I have about this manuscript is the choice to analyze only the months of AMJ. Yes, this is an active period of MCSs (especially those associated with baroclinic waves as noted by the authors), but especially in the region where the trends in MCS precipitation are shown to be large (in Fig. 1b), heavy MCS precipitation also occurs frequently in July and August, and it's not clear to me why these months were not included in the analysis. (If I were to design a study looking at trends in MCS rainfall in the central US, I'd probably select MJJA as the months to analyze. Also keep in mind that the "flood of record" in much of the central US was caused in large part by MCS rainfall in July of 1993, and that event is thus excluded from this analysis.)

As a result, I think one could raise an alternate hypothesis to the one presented in this manuscript: that MCSs in the upper Midwest that previously occurred frequently in July are now starting to occur more regularly in June. (Either instead of, or in addition to, July.) This could be interpreted either as a change in seasonality of heavily raining MCSs, or a poleward shift in where heavily raining MCSs occur at a particular time of year. Of course, these arguments are only subtly different from the arguments that the authors are making. Both are related to changes brought about by climate change, but the resulting interpretations do differ, from "MCSs are producing more rainfall as a result of climate change" vs. "July-like MCSs in Iowa are now happening more frequently in May and June". Thus, I feel that without considering the summertime MCS data, this manuscript isn't telling the whole story about how MCS rainfall might be changing. It might be that running the same analysis for AMJJA gives the same general results, in which case the arguments made in the current manuscript would be that much stronger. Or it could be that there are other changes in seasonality of MCSs that would be revealed in that analysis that would more fully represent the reasons for changes in the climatology of heavy rainfall.

Minor comments:

1. Lines 85-88: I think the analysis of trends from 1997 onward analysis is somewhat problematic. First, this is a relatively short record (less than 20 years) to be using in attribution of climate signals. Second, it appears that 1997 had possibly the highest % of non-MCS precip amounts (Fig. 2c), and one of the lowest MCS frequencies (Fig. 2e), so the trends for at least these variables are being amplified because of starting points that are low in the context of the full record. I do think it's important to indicate that there are two slightly different datasets before and after 1997, but I'm not sure it's warranted to calculate the trends separately, or perhaps the authors can conduct some additional statistical tests to confirm the robustness of these short-term trends.
2. Line 107: Based on the figure legend, these are 5-year periods, not 3-year periods.
3. Lines 128-133: It's not totally clear to me how the calculation of changes in environmental variables is down when there are (apparently) more HP-MCSs in the later years? You're calculating trends in environmental variables but with a changing sample size over time, and possibly also changing locations...do these differences matter to the interpretation of the results? One way to address this might be to display a map of all the HP-MCS locations (perhaps color-coded by the 5-year periods like in the other figures), as well as a time series. This would help to clarify whether the changes in the environments are local changes (e.g., the LLJ for MCSs in Iowa is now on average stronger than in the past) or spatial shifts (there are more MCSs in Iowa than in Missouri now, which is why there are corresponding changes in the LLJ.) This could go in the supplemental

material or the main text depending on the authors' preferences.

4. Lines 154-159: Related to my major comment, I would think that all of these changes are relevant in July and August too (other than the baroclinic waves).

5. Figure 2 caption: The caption needs to state somewhere that these calculations aren't for the entire year, but for a subset of months (AMJ in the current analysis, or whatever it might be after revisions.)

6. Figure 4 caption: This caption needs to indicate that this is for HP-MCSs. A reader could easily be misled to think that these maps are for the mean climate rather than for a set of specific events.

Reviewer #3 (Remarks to the Author):

Review for "More frequent intense and long-lived storms dominate the trend in central U.S. rainfall" by Z. Feng, L. R. Leung, S. Hagos, R. A. Houze, C. D. Burleyson, and K. Balaguru

The authors examined the changes of Mesoscale Convective Systems (MCSs) identified from the North American Land Data Assimilation System (NLDAS) and National Climatic Data Center (NCDC) hourly precipitation products and found that the intensity, lifetime, and frequency of the MCSs over the central U.S. increased from 1979-2014 in spring (April-June). These increases are attributed to the warming over the Rockies which increases the near surface pressure gradient over central U.S. and thus intensifies the Great Plains low-level jet and its northward moisture transport, favoring the formation of long-lived MCSs. For the first time, this study clarifies that the observed increase of extreme precipitation in the central U.S. is largely due to the increase of MCSs. The results are clearly presented and the paper is well written. My comments and questions are listed below for the authors to consider.

1. Line 76, an increase of MCS rainfall by a rate of 20-40% per decade is very large. I wonder if you have examined the trend of total daily precipitation (such as shown in Fig 2a) in other precipitation datasets, and whether the results are similar.

2. Lines 106-107, isn't it 36 years for the period of 1979-2014?

3. Line 107, why use 3-year period? And it seems not consistent with the 5-year period used in Fig. 3d. Could you add a few lines to explain?

4. Lines 127-128, "...mainly occur in these long-lasting MCSs (Figure 2d)..." seems not shown in Fig. 2d.

5. Lines 125-126, "... (HP-MCSs), which we define as MCSs lasting longer than 8 h and have an accumulated rainfall..." is the HP-MCSs calculated from the NARR 3-hourly precipitation? If so, how 8-hourly rainfall is derived from 3-hourly analysis? What's the percent of HP-MCSs in total MCSs?

6. Line 156, add a few more references such as Higgins et al. 1997?

7. Line 159, Fig. 4c instead of 3c?

8. Lines 165-173 state that the intensified GPLLJ transports more moisture into the northern Great Plains and facilitates the development of the MCSs. Why does the increased moisture transport not support an increase of non-MCSs, as increased low-level moisture tend to destabilize the atmosphere? In other words, why do non-MCSs decrease? Is it because non-MCSs are transformed to MCSs?

9. Fig. 4, hashes are somehow hard to see. Could you modify the figure to make them clearer?

The unit of moisture flux in Fig. 4c seems not correct. Is it the trend of qv at 850 hPa or the trend of vertically integrated moisture flux? And could you add mask for the topography below 850 hPa in Fig. 4c? Also, I wonder if it is better to show the trend of 850 hPa meridional winds in Fig. 4d instead of specific humidity. The increase of moisture flux may be due to either the increase of the moisture or the meridional wind or both of them, and it may be better to show that the jet is intensified here instead of in the supplementary figures.

10. Figs. S1c and d, green hashes are hard to see under purple shading. Could you adjust the color?

11. Please add mask for topography for Figs. S1, S2c-d. The unit of vectors in Fig. S2d seems not correct as well.

12. Fig. S3 is a good example showing that PF defined MCSs are quite similar to those defined by cold cloud shield. It would be good to add a few lines to explain why these two criteria are exchangeable.

13. Fig. S4, "Mean values are shown by the solid red lines" seems not clear in the figure.

14. Lines 190-201, hourly weight is used to create hourly NLDAS precipitation, right? If so, does the weight vary from year to year? If not, the changes of hourly precipitation largely reflect the variations of the daily precipitation, correct?

Reference

Higgins, R. W., Y. Yao, E. S. Yarosh, J. E. Janowiak, and K. C. Mo (1997), Influence of the Great Plains low-level jet on summertime precipitation and moisture transport over the central United States, *J. Clim.*, 10(3), 481-507, doi:10.1175/1520-0442(1997)010<0481:Iotgpl>2.0.Co;2.

We thank all three reviewers for providing constructive comments and suggestions for improving the manuscript. We made our best attempt to address the comments. Please see our point-by-point response (in blue) in detail below.

Reviewer #1 (Remarks to the Author):

I have the following three major comments about the manuscript

1. I find the methods description in the supplementary information somewhat confusing and inadequate. I should be able to reproduce their analysis with this information. There is critical detail missing that prevents a full understanding. For example, I am not sure of the meaning of the least squares fit lines in Fig. S4. What is being regressed against what? It certainly is not a fit to the other data being displayed in that figure. Is this a fit to a particular percentile value for each duration bin?

We added additional discussion in deriving the PF parameters used to identify MCS and made substantial revision from pages 5 to 8 in the supplementary information, particularly clarifying the regression procedures in Figure S5 (previously Figure S4).

To answer your question regarding the regression: the fit lines are indeed regression between lifetime (x-axis) and a particular percentile value of the three parameters (y-axis) in Figure S5. Essentially, these three PF parameters are additional lifetime-dependent thresholds to identify MCS besides the minimum size and duration criteria. We added description of how the final fit lines in Figure S5 were determined, and added an example of how the skill score is calculated (Figure S6).

These three PF parameters are then adjusted following the procedures described in detail on page 11 to account for the changes introduced in the NLDAS dataset in 1996. The adjusted fit lines for data prior to 1996 are shown as black lines in Figure S10. In general, for NLDAS data prior to 1996, the MCS definition is adjusted to have larger maximum areal extent, smaller maximum areal mean rain-rate, and similar maximum pixel-level rain-rate skewness compared to data after 1996.

2. The differences in PF characteristics around the change point of 1996 are surprisingly large (of the order of a factor of 2) and somewhat alarming, raising questions about the adequacy of the sub-daily precipitation. I have concerns about the validity of use of this dataset, even if adjustments are made, particularly in regard to trend analysis. These differences are quite large compared to the trend magnitudes.

The reviewer has raised an important point. To address this comment, we repeated the trend analysis using an independent dataset from NCDC rain gauges (see Figure R1 below). The calculation is carried out as follows:

- For each of the NCDC rain gauge station data within the central U.S. (a total of 261 stations in the magenta box in Fig. 1), the recorded 1-hourly precipitation amount is classified into either “MCS” or “non-MCS” using the NLDAS 1-hourly MCS location maps
- The 1-hourly total, MCS, and non-MCS precipitation amount is accumulated into seasonal total amount, then divided by the total number of hours to convert the unit to [mm day⁻¹]
- Precipitation frequency is calculated by dividing the number of hours with NCDC hourly precipitation > 0 mm by the total number of NCDC sampling hours (including no precipitation) in a season.

The resulting trend analysis using NCDC rain gauge data is qualitatively consistent with the NLDAS dataset. The magnitudes of the trends for both MCS and non-MCS precipitation amount and frequency from NCDC data are larger than NLDAS, possibly due to differences between point measurements for the NCDC rain gauge dataset and the spatially interpolated NLDAS dataset at 1/8-degree resolution.

These results suggest that when NLDAS hourly data are aggregated to daily resolution, the impact from incorporating NEXRAD data since 1997 is essentially removed. Therefore, the trends shown in Fig. 2 are robust. We have added several sentences in the paragraph describing the results in Fig. 2 to reflect our response to this comment.

Figure R1. Similar to Figure 2, except using the 1-hourly precipitation from NCDC rain gauges for calculating precipitation amount and frequency. Identification of MCS vs. non-MCS precipitation uses the NLDAS 1-hourly MCS location maps. (a) Total precipitation, (b) MCS precipitation, (c) non-MCS precipitation, (d) total precipitation frequency, (e) MCS precipitation frequency, and (f) non-MCS precipitation frequency.

3. The statement "Up to now, no study has examined the phenomenological aspects of the observed increases" is not true, at least not without some substantial caveats. I led a rather extensive study of the phenomenology of extreme precipitation increases. This is published as

Kunkel, K.E., D.R. Easterling, D.A.R. Kristovich, B. Gleason, L. Stoecker, and R. Smith, 2012: Meteorological causes of the secular variations in observed extreme precipitation events for the conterminous United States. *J. Hydromet.*, 13, 1131-1141.

In the above study, we found that fronts were principally responsible for the upward trends in extreme precipitation events in the central part of the U.S. On the surface, this is a quite different conclusion. The apparent differences between our findings and those of this study probably derive from the following:

Definition of extremes. The authors here are using the 95th percentile of daily rainfall. That is actually not "extreme" and the authors should use another word, such as "heavy". In my study, we used the 5-yr return period threshold, which is more than an order of magnitude more extreme.

Yes we agree that using 95th percentile is not often considered "extreme" in traditional analysis of all rainfall data. But our study differs from traditional analysis in that we first separate rainfall events into MCSs and non-MCSs, which naturally partitions rainfall into heavy vs. light categories. Then we look at the "tail" of the rainfall intensity within MCSs, which are generally heavy rainfall events. Therefore, the results in Fig. 3 are representative of changes in the extreme rainfall intensity.

Definition of MCS. My study extended back to the early part of the 20th Century, well before any satellite information could inform that study. We were restricted by what could be identified from surface temperature and precipitation patterns, daily weather maps, and 20th Century reanalysis (before 1948). We categorized the meteorology according to what is likely the primary forcing. If an MCS-like precipitation feature were along a warm or cold front of an extratropical cyclone, we considered the frontal convergence to be the primary cause, although (of course) convective instability could greatly enhance vertical motion. Our MCS category was narrowly defined to those precipitation features that were likely (initiated and) driven primarily by convective instability, some distance from any ETC front. Our identification was done by expert judgment, i.e. we for the most part did not use automated tools. This was quite laborious and, although potentially subject to certain biases, it gave us an intimate feel for the phenomenology for these situations.

Our trend analysis was done for a 100+-year period. While we found overall upward trends for frontally-caused events, a substantial portion of that came from increases prior to 1979. Also, our trend analysis was annual. We did not perform a seasonal trend analysis.

Thank you for pointing out this interesting study. We have added the reference of Kunkel et al. (2012) to the revision, with a discussion of the difficulties in identifying MCSs associated with frontal systems to be consistent with post-satellite era studies. Such difficulties preclude a separation of MCS versus non-MCS rainfall events to determine and understand their respective changes, which is the main goal of our study.

Reviewer #2 (Remarks to the Author):

General comments:

In this study, the authors analyze rainfall data for the late spring (April, May, June; hereinafter AMJ) in the central United States, and aim to identify trends over the period 1979-2014 in the amounts of rainfall produced by mesoscale convective systems (MCSs) in this region. Based on the analysis, they show that particularly in the upper Midwest (Iowa, Wisconsin, etc.) there has been a large increase in MCS-produced rainfall over this time period and that, in turn, this is associated with a stronger Great Plains Low-Level Jet (GPLLJ) during MCSs. The data and analysis methods are generally sound and the manuscript is very clearly written and presented. However, I have one fairly substantive question/comment that may potentially affect the interpretation of the results that I believe the authors should address prior to the manuscript being ready for publication, along with a few minor suggested revisions.

Major comment:

The primary question/concern I have about this manuscript is the choice to analyze only the months of AMJ. Yes, this is an active period of MCSs (especially those associated with baroclinic waves as noted by the authors), but especially in the region where the trends in MCS precipitation are shown to be large (in Fig. 1b), heavy MCS precipitation also occurs frequently in July and August, and it's not clear to me why these months were not included in the analysis. (If I were to design a study looking at trends in MCS rainfall in the central US, I'd probably select MJJA as the months to analyze. Also keep in mind that the "flood of record" in much of the central US was caused in large part by MCS rainfall in July of 1993, and that event is thus excluded from this analysis.)

As a result, I think one could raise an alternate hypothesis to the one presented in this manuscript: that MCSs in the upper Midwest that previously occurred frequently in July are now starting to occur more regularly in June. (Either instead of, or in addition to, July.) This could be interpreted either as a change in seasonality of heavily raining MCSs, or a poleward shift in where heavily raining MCSs occur at a particular time of year. Of course, these arguments are only subtly different from the arguments that the authors are making. Both are related to changes brought about by climate change, but the resulting interpretations do differ, from "MCSs are producing more rainfall as a result of climate change" vs. "July-like MCSs in Iowa are now happening more frequently in May and June". Thus, I feel that without considering the summertime MCS data, this manuscript isn't telling the whole story about how MCS rainfall might be changing. It might be that running the same analysis for AMJJA gives the same general results, in which case the arguments made in the current manuscript would be that much stronger. Or it could be that there are other changes in seasonality of MCSs that would be revealed in that analysis that would more fully represent the reasons for changes in the climatology of heavy rainfall.

Thank you for raising an important point about an alternative hypothesis for MCS changes. We have added the summer MCS rainfall trend map in the Supplemental Information (Figure S1). The results show that a majority of the MCS total rainfall increase in summer occurs in the Southern Great Plains, parts of the Midwest and Great Lakes region, and southeastern U.S. Large areas in the central Great Plains where maximum climatological MCS rainfall occurs do not show statistically significant trends. Therefore, we think that the mechanisms for summer MCS changes likely differ from those for the late spring. The results do not provide evident support for a seasonality shift, as there is no decreasing trend in MCS rainfall in the Central and Northern Great Plains in the summer months that is complementary to the changes in spring. The causes for the change in summer MCS precipitation are being investigated for a future study. We added these discussions in the paragraph describing Figure 1 in the main text.

Minor comments:

1. Lines 85-88: I think the analysis of trends from 1997 onward analysis is somewhat problematic. First, this is a relatively short record (less than 20 years) to be using in attribution of climate signals. Second, it appears that 1997 had possibly the highest % of non-MCS precip amounts (Fig. 2c), and one of the lowest MCS frequencies (Fig. 2e), so the trends for at least these variables are being amplified because of starting points that are low in the context of the full record. I do think it's important to indicate that there are two slightly different datasets before and after 1997, but I'm not sure it's warranted to calculate the trends separately, or perhaps the authors can conduct some additional statistical tests to confirm the robustness of these short-term trends.

We agree that the period from 1997 to current (18 years) is relatively short for trend analysis. We did a trend analysis for the recent decade because of the change in the NLDAS dataset as we stated in the Supplemental Information. The NLDAS data since 1997 incorporates the NEXRAD radar network data to improve the spatial and temporal representation of their precipitation product. Therefore, our intention for a trend analysis for the second period is to show that the trend in the recent decade obtained using radar era NLDAS data is qualitatively consistent with the long-term trend we found using the full NLDAS record, suggesting that the long-term trend is not affected by the change in the dataset or our analysis method.

We tested excluding 1997 in the regression for the recent decade, and it did not affect the magnitude of the trends and their statistical significance (see Figure R2).

Figure R2. Same as Figure 2 except recent decade trend is calculated using 1998-2014 data (excluding 1997).

2. Line 107: Based on the figure legend, these are 5-year periods, not 3-year periods. We apologize for the mistake. It has been changed to 5-year period now for consistency. We have changed to show the 95th percentile exceedance frequency and 95th percentile rain-rate trend for MCSs to better represent “extremes”. We also fixed a bug in our calculation for the exceedance frequency trend calculation in Figure 3a,c, resulting in more sites with statistically significant trends in the intensity. Figure 3 has been updated accordingly.

3. Lines 128-133: It's not totally clear to me how the calculation of changes in environmental variables is down when there are (apparently) more HP-MCSs in the later years? You're calculating trends in environmental variables but with a changing sample size over time, and possibly also changing locations...do these differences matter to the interpretation of the results? One way to address this might be to display a map of all the HP-MCS locations (perhaps color-coded by the 5-year periods like in the other figures), as well as a time series. This would help to clarify whether the changes in the environments are local changes (e.g., the LLJ for MCSs in Iowa is now on average stronger than in the past) or spatial shifts (there are more MCSs in Iowa than in Missouri now, which is why there are corresponding changes in the LLJ.) This could go in the supplemental material or the main text depending on the authors' preferences.

The number of MCSs in our analysis is rather large in every season. The average number of HP-MCSs in each season is 92 and the lowest number is 50. With the minimum lifetime of 8 hours (2-3 NARR snapshots) for each HP-MCS, there are on average 152 snapshots of NARR large-scale environments

used in the composite for each season (minimum of 62 snapshots). Therefore, the seasonal composites are representative of the regional circulation pattern when MCSs occur. A sentence has been added in this paragraph in the revision.

We also note that a majority of the changes in total MCS rainfall is associated with long-lasting (lifetime > 8 h) MCSs (see Figure R3 below). HP-MCSs are just a further subset of these long-lasting MCSs, which account for ~80% of the MCS rainfall (see Figure R6 and response to Reviewer 3's comment #5).

Figure R3. Same as Figure 1 except for long-lasting (> 8 h) MCSs.

4. Lines 154-159: Related to my major comment, I would think that all of these changes are relevant in July and August too (other than the baroclinic waves). Yes the GPLLJ also affects summer time MCS precipitation. However, as shown in Figure S1, the changes in summer MCS clearly show a different pattern compared to late spring, suggesting the underlying mechanisms for the change are different between the two seasons.

5. Figure 2 caption: The caption needs to state somewhere that these calculations aren't for the entire year, but for a subset of months (AMJ in the current analysis, or whatever it might be after revisions.)
"April-June" has been added to the caption.

6. Figure 4 caption: This caption needs to indicate that this is for HP-MCSs. A reader could easily be misled to think that these maps are for the mean climate rather than for a set of specific events.
"Surface and 850 hPa trends during occurrence of HP-MCSs." has been added to the caption.

Reviewer #3 (Remarks to the Author):

Review for "More frequent intense and long-lived storms dominate the trend in central U.S. rainfall" by Z. Feng, L. R. Leung, S. Hagos, R. A. Houze, C. D. Burleyson, and K. Balaguru

The authors examined the changes of Mesoscale Convective Systems (MCSs) identified from the North American Land Data Assimilation System (NLDAS) and National Climatic Data Center (NCDC) hourly precipitation products and found that the intensity, lifetime, and frequency of the MCSs over the central U.S. increased from 1979-2014 in spring (April-June). These increases are attributed to the warming over the Rockies which increases the near surface pressure gradient over central U.S. and thus intensifies the Great Plains low-level jet and its northward moisture transport, favoring the formation of long-lived MCSs. For the first time, this study clarifies that the observed increase of extreme precipitation in the central U.S. is largely due to the increase of MCSs. The results are clearly presented and the paper is well written. My comments and questions are listed below for the authors to consider.

1. Line 76, an increase of MCS rainfall by a rate of 20-40% per decade is very large. I wonder if you have examined the trend of total daily precipitation (such as shown in Fig 2a) in other precipitation datasets, and whether the results are similar.

Reviewer 1 asked a similar and important question. The following response is the same as the response to Reviewer 1's comment #2.

We repeated the trend analysis using an independent dataset from NCDC rain gauges (see Figure R4 below). The calculation is carried out as follows:

- For each of the NCDC rain gauge station data within the central U.S. (a total of 261 stations in the magenta box in Fig. 1), the recorded 1-hourly precipitation amount is classified into either "MCS" or "non-MCS" using the NLDAS 1-hourly MCS location maps
- The 1-hourly total, MCS, and non-MCS precipitation amount is accumulated into seasonal total amount, then divided by the total number of hours to convert the unit to [mm day⁻¹]
- Precipitation frequency is calculated by dividing the number of hours with NCDC hourly precipitation > 0 mm by the total number of NCDC sampling hours (including no precipitation) in a season.

The resulting trend analysis using NCDC rain gauge data is qualitatively consistent with the NLDAS dataset. The magnitudes of the trends for both MCS and non-MCS precipitation amount and frequency from NCDC data are larger than NLDAS, possibly due to differences between point measurements for the NCDC rain gauge dataset and the spatially interpolated NLDAS dataset at 1/8-degree resolution.

These results suggest that when NLDAS hourly data are aggregated to daily resolution, the impact from incorporating NEXRAD data since 1997 is essentially removed. Therefore, the trends shown in Fig. 2 are robust. We have added several sentences in the paragraph describing the results in Fig. 2 to reflect our response to this comment.

Figure R4. Similar to Figure 2, except using the 1-hourly precipitation from NCDC rain gauges for calculating precipitation amount and frequency. Identification of MCS vs. non-MCS uses NLDAS 1-hourly MCS location map. (a) total precipitation, (b) MCS precipitation, (c) non-MCS precipitation, (d) total precipitation frequency, (e) MCS precipitation frequency, and (f) non-MCS precipitation frequency.

2. Lines 106-107, isn't it 36 years for the period of 1979-2014?

Yes, thank you for pointing that out. It has been changed to 36 years now throughout the paper.

3. Line 107, why use 3-year period? And it seems not consistent with the 5-year period used in Fig. 3d. Could you add a few lines to explain?

We apologize for the mistake. It has been changed to 5-year period now for consistency. We have changed to show the 95th percentile exceedance frequency and 95th percentile rain-rate trend for MCSs to better represent “extremes”. We also fixed a bug in our calculation for the exceedance frequency trend calculation in Figure 3a,c, resulting in more sites with statistically significant trends in the intensity. Figure 3 has been updated accordingly.

4. Lines 127-128, "...mainly occur in these long-lasting MCSs (Figure 2d)..." seems not shown in Fig. 2d.

We inspected the trend of the 95th percentile MCS lifetime for long-lasting (> 8 h) MCSs and found a larger trend of 7% decade⁻¹ (see Figure R5). These long-lasting MCSs are the main reason driving the mean MCS lifetime trends. The sentence has been revised: "The increase is particularly striking in the frequency of very long-lasting MCSs, where the trend of the 95th percentile MCS lifetime is ~7% decade⁻¹."

Figure R5. Trends in long-lasting MCS (lifetime > 8 h). Magenta line is the fit for the 95th percentile MCS lifetime values.

5. Lines 125-126, "...(HP-MCSs), which we define as MCSs lasting longer than 8 h and have an accumulated rainfall..." is the HP-MCSs calculated from the NARR 3-hourly precipitation? If so, how 8-hourly rainfall is derived from 3-hourly analysis? What's the percent of HP-MCSs in total MCSs?

The HP-MCSs are defined using the NLDAS dataset, not NARR. Assuming the reviewer is asking about the "percentage of rainfall" from HP-MCSs to total MCSs, HP-MCSs account for about 80% of the total MCS rainfall (see Figure R6).

Figure R6. (a) Total accumulated rainfall per MCS as a function of MCS lifetime. Boxes show the 25th and 75th percentiles and the dark blue bars show the median values. The red line shows least square fit to the median values. Black squares are the total number of MCS per lifetime and red squares are the number of HP-MCSs. (b) Total MCS rainfall as a function of MCS lifetime factoring in the total number of MCSs at each lifetime bin. The blue bars show all MCSs and the red bars are for HP-MCSs. The black and red lines denote the cumulative distribution function of all MCS and HP-MCS rainfall, respectively.

6. Line 156, add a few more references such as Higgins et al. 1997?

The influence of GPLLJ to enhanced precipitation in the Great Plains has been added with the Higgins et al. 1997 reference.

7. Line 159, Fig. 4c instead of 3c?

Yes it should be Figure 4c, corrected.

8. Lines 165-173 state that the intensified GPLLJ transports more moisture into the northern Great Plains and facilitates the development of the MCSs. Why does the increased moisture transport not support an increase of non-MCSs, as increased low-level moisture tend to destabilize the atmosphere? In other words, why do non-MCSs decrease? Is it because non-MCSs are transformed to MCSs?

The increased strength of the GPLLJ and associated moisture transport is stronger during nighttime than daytime (Figure S2), and MCS activity in the Great Plains peak at nighttime, while non-MCS precipitation (e.g. local convective storms) maximize in the afternoon. Therefore, the enhanced GPLLJ would preferentially favor increased frequency and intensity of nocturnal MCSs rather than daytime local convection.

The reviewer's suggestion about whether non-MCSs are transformed to MCS is an interesting possibility. Exactly why and how isolated convection grows upscale or aggregates to become MCS are still research questions actively being pursued in the convection community. We currently do not have concrete evidence to support this possibility, so we prefer not to speculate on this.

9. Fig. 4, hashes are somehow hard to see. Could you modify the figure to make them clearer? The unit of moisture flux in Fig. 4c seems not correct. Is it the trend of q_v at 850 hPa or the trend of vertically integrated moisture flux? And could you add mask for the topography below 850 hPa in Fig. 4c? Also, I wonder if it is better to show the trend of 850 hPa meridional winds in Fig. 4d instead of specific humidity. The increase of moisture flux may be due to either the increase of the moisture or the meridional wind or both of them, and it may be better to show that the jet is intensified here instead of in the supplementary figures.

Fig. 4 has been updated to address the reviewer's suggestion: 1) terrain height is added to Fig. 4b, 2) 850 hPa moisture flux vectors are replaced with wind vectors in Fig. 4c,d, 3) hashes are thickened to make it easier to see.

The moisture flux is calculated using $Flux = \frac{1}{g} \times V \times q_v \times dp$, where dp is the pressure layer thickness between 850 hPa and 825 hPa. Therefore, the unit is in $kg\ m^{-1}\ s^{-1}$.

10. Figs. S1c and d, green hashes are hard to see under purple shading. Could you adjust the color?

Hashes in Figure. S2c,d have been thickened.

11. Please add mask for topography for Figs. S1, S2c-d. The unit of vectors in Fig. S2d seem not correct as well.

Terrain contours have been added to Fig. S2a,b. and Fig. S3b. The vectors are wind speed in this figure.

12. Fig. S3 is a good example showing that PF defined MCSs are quite similar to those defined by cold cloud shield. It would be good to add a few lines to explain why these two criteria are exchangeable.

The following two sentences have been added to page 4 in the Supplementary Information.

“Note that the PF and cold cloud shield associated with the MCS have reasonable correspondence in the spatial and temporal evolution throughout most of the life cycle of the MCS. Such correspondence has been explored in previous works using a combination of satellite-based cold cloud shields and ground-based radar and/or rain gauge dataset to understand the structure and evolution of MCSs.”

13. Fig. S4, "Mean values are shown by the solid red lines" seems not clear in the figure.

Mean values are shown by solid red line with filled circles, the figure caption has been updated to clarify this.

14. Lines 190-201, hourly weight is used to create hourly NLDAS precipitation, right? If so, does the weight vary from year to year? If not, the changes of hourly precipitation largely reflect the variations of the daily precipitation, correct?

To our understanding, the weightings are dynamically calculated for each grid and each hour, based on available remote sensing platforms after 1996. For example, if Stage II radar hourly precipitation data is available, the radar derived hourly precipitation is divided by the radar daily total to create hourly temporal disaggregation weights. These hourly weights are then multiplied by the daily gauge-only CPC precipitation analysis to obtain the hourly NLDAS precipitation. Before 1996, CPC hourly CONUS data is used without incorporating any remote sensing data. Therefore, the weightings do change with time. More details about the NLDAS dataset are available on the webpage link provided in Methods.

Reference

Higgins, R. W., Y. Yao, E. S. Yarosh, J. E. Janowiak, and K. C. Mo (1997), Influence of the Great Plains low-level jet on summertime precipitation and moisture transport over the central United States, *J. Clim.*, 10(3), 481-507, doi:10.1175/1520-0442(1997)010<0481:lotgpl>2.0.Co;2.

REVIEWERS' COMMENTS:

Reviewer #1 (Remarks to the Author):

The changes to the manuscript address my concerns.

I found one additional minor error. The panels in Fig. 3 indicate that the analysis is for the 95th percentile events; this is also indicated in the text. However, the caption says 90th percentile.

Reviewer #2 (Remarks to the Author):

For the most part, the authors have convincingly responded to the questions and comments I raised during the first review. However, I do have a couple areas in the paper that I think need to be either revised, or clarified further, prior to publication; along with a few minor editorial corrections.

Specific comments:

1. I appreciate the authors' presentation of the analogous calculations for summer in their response, which is quite interesting, essentially revealing that there is no summer trend in MCS precipitation in the region where MCSs are frequent in the summer. However, because those results indicate that the results presented in this paper do not apply to summer MCSs, I suggest adding the season to the article title so that it reads "More frequent intense and long-lived storms dominate the *late-spring* trend in central US rainfall". Because the results have been shown to only apply in this season, I think this qualifier is important.
2. Lines 54-56: based on the comment of reviewer 1, I think this sentence should be modified to say that "Extreme precipitation is defined *here* as..." since your definition is less "extreme" than those typically used in the literature.
3. Line 84: revise "increase" to "increased"
4. Line 154: suggest adding the caveat that these results only apply to late spring...for example, add clause ", at least in late spring" to the end of the sentence, or something similar.
5. Section on environment changes: I'm still confused by a few things in this section, which I think are important to clarify. First, I don't understand how the frequency of HP-MCSs can be as high as reported here. If the average number per season is 91, that means there is one of these MCSs per day in April-June -- on average! Which then means that, since these are defined as having a precipitation accumulation greater than the median for all of these long-lived MCSs that there must be at least two 8-hour-long MCSs every day (one above the median for rainfall, and one below). (And some years must have substantially more than this.) Of course MCSs are frequent in this region, but I just don't think there's an average of two 8-hour duration MCSs, every day in the late spring. So I think some clarifying info, or maybe a map of the density of these HP MCSs as I suggested earlier, might be warranted in the supplement.

Then, in the discussion of the environmental results (specifically lines 179-203), I think more care needs to be taken with the wording. The way that these paragraphs currently read make it sound like the changes you show are changes to the overall climate, but what you've really shown are only changes that occur on HP-MCS days. In other words, maybe there is a large-scale change in the strength of the subtropical high that increases the mass in the SE US in spring, but that's not what your analysis is showing. (And so on for the moisture changes, LLJ changes, etc.) Your analysis is showing that when heavily raining MCSs occur now, the subtropical high is stronger than it used to be on HP-MCS days in the past. I think this distinction is important but it isn't really reflected in this discussion. This section should be revised to make these paragraphs indicate that the changes you're showing are changes on HP-MCS days, not changes to the overall springtime climate. (Of course, the two are likely related, and I think it's ok to make some speculation that they are related, but you haven't actually shown that relationship here. And I guess if it's really true that the sample includes nearly all days in the record, then the two would be effectively equivalent, but I'm still not convinced of this, as noted above.)

6. Figure 3 caption: The caption says 90th percentile, but the figure shows 95th percentile. Make sure to correct this.
7. Supplement, Figure S8: the caption here needs to indicate that the analysis is for April-June rather than the entire year.

Reviewer #3 (Remarks to the Author):

The authors did a great job in revising the paper. Most of my comments and suggestions are clearly addressed. One concern I have is their reply to my previous comments on Figures 4c (not Fig. 4b), S2a-b, and S3b. I probably did not address my question clearly but I suggested masking out the topography below 850 hPa in those figures instead of adding terrain height. For regions with surface pressure greater than 850 hPa, such as the Rockies, values at 850 hPa pressure level are artificially interpolated. This is only my suggestion for the authors to consider, and they can keep the figures as they were in the previous version.

Another minor comment is in line 201, I think "850 hPa moisture and circulation" are referred to "Figure 4d" instead of "Figure 3d".

We thank the reviewers again for providing additional comments for further improving the manuscript. Please see our point-by-point response (in blue) in detail below.

Reviewer #1 (Remarks to the Author):

The changes to the manuscript address my concerns.

I found one additional minor error. The panels in Fig. 3 indicate that the analysis is for the 95th percentile events; this is also indicated in the text. However, the caption says 90th percentile.

The captions have been corrected to 95th percentile.

Reviewer #2 (Remarks to the Author):

For the most part, the authors have convincingly responded to the questions and comments I raised during the first review. However, I do have a couple areas in the paper that I think need to be either revised, or clarified further, prior to publication; along with a few minor editorial corrections.

Specific comments:

1. I appreciate the authors' presentation of the analogous calculations for summer in their response, which is quite interesting, essentially revealing that there is no summer trend in MCS precipitation in the region where MCSs are frequent in the summer. However, because those results indicate that the results presented in this paper do not apply to summer MCSs, I suggest adding the season to the article title so that it reads "More frequent intense and long-lived storms dominate the *late-spring* trend in central US rainfall". Because the results have been shown to only apply in this season, I think this qualifier is important.

We added "springtime" in the title as April-June includes most of spring.

2. Lines 54-56: based on the comment of reviewer 1, I think this sentence should be modified to say that "Extreme precipitation is defined *here* as..." since your definition is less "extreme" than those typically used in the literature.

Done.

3. Line 84: revise "increase" to "increased"

Corrected.

4. Line 154: suggest adding the caveat that these results only apply to late spring...for example, add clause ", at least in late spring" to the end of the sentence, or something similar.

We added "at least during the spring season" at the end of the sentence.

5. Section on environment changes: I'm still confused by a few things in this section, which I think are important to clarify. First, I don't understand how the frequency of HP-MCSs can be as high as reported here. If the average number per season is 91, that means there is one of these MCSs per day in April-June -- on average! Which then means that, since these are defined as having a precipitation accumulation greater than the median for all of these long-lived MCSs that there must be at least two 8-hour-long MCSs every day (one above the median for rainfall, and one below). (And some years must have substantially more than this.) Of course MCSs are frequent in this region, but I just don't think there's an average of two 8-hour duration MCSs, every day in the late spring. So I think some clarifying info, or maybe a map of the density of these HP MCSs as I suggested earlier, might be warranted in the supplement.

Our methodology tracks Precipitation Features (PF) and uses that as a proxy to identify MCSs. Compared to identifying MCSs using satellite cold cloud shield, which is the traditional method for the satellite era, the cloud shield tend to be much

larger and could contain multiple PFs underneath. In the satellite definition, all PFs underneath the tracked cold cloud shield are considered belonging to one MCS. But in the PF definition, they could be treated as several MCSs. Therefore, the absolute number of MCSs is not that important, as it would depend on the dataset used to define MCS.

An example of an extremely large MCS identified by satellite cold cloud shield is given in the figure below. The cloud shield covering the entire central U.S. from eastern Texas to the Great Lakes is treated as one MCS owing to contiguous $T_b < 240$ K. But underneath the cold cloud shield, up to three distinct PFs can be seen for this event. Using our PF method, these are treated as three distinct MCSs. This example highlights the absolute number of MCSs is not that important, as long as precipitation associated with mesoscale convection is properly captured by our methodology.

Figure R1. Example of a Mesoscale Convective System (MCS) tracked by satellite cold cloud shield containing multiple Precipitation Features (PFs).

Then, in the discussion of the environmental results (specifically lines 179-203), I think more care needs to be taken with the wording. The way that these paragraphs currently read make it sound like the changes you show are changes to the overall climate, but what you've really shown are only changes that occur on HP-MCS days. In other words, maybe there is a large-scale change in the strength of the subtropical high that increases the mass in the SE US in spring, but that's not what your analysis is showing. (And so on for the moisture changes, LLJ changes, etc.) Your analysis is showing that when heavily raining MCSs occur now, the subtropical high is stronger than it used to be on HP-MCS days in the past. I think this distinction is important but it isn't really reflected in this discussion. This section should be revised to make these paragraphs indicate that the changes you're showing are changes on HP-MCS days, not changes to the overall springtime climate. (Of course, the two are likely related, and I think it's ok to make some speculation that they are related, but you haven't actually shown that relationship here. And I guess if it's really true that the sample includes nearly all days in the record, then the two would be effectively equivalent, but I'm still not convinced of this, as noted above.)

Thank you for your suggestion to look at the climatological trends. We repeated our analysis using all times during April-June to examine the climatological large-scale trends in spring (see two figures below). The results show that the spatial pattern of the climatological trends is consistent with those composited during HP-MCSs, although the magnitudes of the climatological trend are weaker. We think that the climatological large-scale environment is indeed changing, as the reviewer suggested. The weaker climatological trends are likely caused by averaging conditions that are not favorable for MCSs, which further warrant our strategy to focus on environments during HP-MCS days. A sentence is added at the end of the **“Role of large-scale environment changes”** section to summarize this point.

Figure R2. Same as Figure 4 except all times during April–June are included in the composite and trend calculation. Note that smaller contour intervals for the MSLP trends in (a) and geopotential height trends in (d) are used here.

Figure R3. Same as Supplementary Figure 2 except all times during April–June are included in the composite and trend calculation. Note that smaller contour intervals for geopotential height trends in (c–d) are used here.

6. Figure 3 caption: The caption says 90th percentile, but the figure shows 95th percentile. Make sure to correct this.
Thank you for pointing that out. It has been corrected.
7. Supplement, Figure S8: the caption here needs to indicate that the analysis is for April-June rather than the entire year.
The figure actually includes all warm season months. “April-August” has been added to the figure caption.

Reviewer #3 (Remarks to the Author):

The authors did a great job in revising the paper. Most of my comments and suggestions are clearly addressed. One concern I have is their reply to my previous comments on Figures 4c (not Fig. 4b), S2a-b, and S3b. I probably did not address my question clearly but I suggested masking out the topography below 850 hPa in those figures instead of adding terrain height. For regions with surface pressure greater than 850 hPa, such as the Rockies, values at 850 hPa pressure level are artificially interpolated. This is only my suggestion for the authors to consider, and they can keep the figures as they were in the previous version.

Thank you for clarifying that. We have masked out those area in Figure 4c-d, Supplementary Figure 2, 3c-d.

Another minor comment is in line 201, I think "850 hPa moisture and circulation" are referred to "Figure 4d" instead of "Figure 3d".

Thank you. It has been corrected.

Reviewer #2:

I thank the authors for their helpful responses and the additional context they provided. I like the comparison between the HP-MCS days and the full April-June climatology, and I'd actually encourage the authors to include Figure R2 in the supplementary information as I think it's something that readers of the paper might be interested in.

The last suggestion I have, based on this same line of discussion, is in regards to the abstract. The new text in the article itself about the role of climatological vs. HP-MCS days is appropriately measured, but the last sentence of the abstract is maybe more definitive than it should be. I think the moisture transport is important to mention as well, as it's not really possible to separate the effects of the LLJ winds (enhanced convergence, inflow, etc.) and the moisture in terms of what's really driving the greater precip. Here's a try at a sentence that might convey all of this:

(current text) "A strengthening of the climatological southerly low-level jet in the Central/Northern Great Plains accounts for the changes in Mesoscale Convective System precipitation."

(possible revised text) "A strengthening of the southerly low-level jet and its associated moisture transport in the Central/Northern Great Plains, in the overall climatology and in particular on days with mesoscale convective systems, accounts for the changes in MCS precipitation."

Though this sentence is a little long, so you might be able to come up with something better! Lastly, a very minor point, but I'm not sure that "mesoscale convective system" needs to have all of the words capitalized in the abstract. It's not really a proper name, but perhaps the Nature guidelines say otherwise.